# Enhanced RF Energy Harvesting System Utilizing Piezoelectric Transformer

**DOI:** 10.3390/s24227111

**Published:** 2024-11-05

**Authors:** Mahmoud Al Ahmad, K. S. Phani Kiranmai, Abdulla Alnuaimi, Obaid Alyammahi, Hamad Alkaabi, Saeed Alnasri, Abdulrahman Dahir

**Affiliations:** Electrical Engineering Department, United Arab Emirates University, Al Ain 15551, United Arab Emirates; 700038622@uaeu.ac.ae (K.S.P.K.); 202005575@uaeu.ac.ae (A.A.); 202000072@uaeu.ac.ae (O.A.); 202011529@uaeu.ac.ae (H.A.); 202007882@uaeu.ac.ae (S.A.); abd.daher@uaeu.ac.ae (A.D.)

**Keywords:** energy harvesting, piezoelectric materials, piezoelectric transformer, RF energy, sustainability

## Abstract

RF energy harvesting converts ambient signals into electrical power, providing a sustainable energy source. This study demonstrates the use of a piezoelectric transformer for efficient RF energy harvesting. In this work, a piezoelectric transformer (PT) is employed as a high-gain, efficient inverting amplifier to enhance RF wireless energy harvesting. The PT, composed of lead zirconate titanate (PZT), is placed after the receiving loop antenna, with its output connected to an AC-to-DC converter circuit. Maximum harvested power was observed at the PT’s resonance frequency of 50 kHz, with an optimal load of 40 kΩ. The system, comprising the antenna, transformer, and rectifier circuit, continues to resonate at 50 kHz, as confirmed by input impedance measurements, demonstrating stable and effective performance. The overall system efficiency was characterized to be 88%.

## 1. Introduction

Wireless communication networks are anticipated to offer enhanced capabilities with more frequencies, expanded bandwidth, and widespread connectivity [1]. This advancement not only promises improved communications but also paves the way for novel applications in energy harvesting technologies [2]. The incorporation of RF energy harvesting into these infrastructures has the potential to create communication systems that are both more sustainable and energy-efficient [3]. Notably, ongoing research and development efforts have primarily focused on two critical areas: the design of individual components and the advancement of power management strategies. Researchers are committed to improving performance by integrating of components within RF harvesting systems. This encompasses optimizing antennas to efficiently capture RF energy across a broad frequency spectrum [4,5,6,7,8]. Significant strides have been made in augmenting the efficiency of rectifiers and implementing innovative voltage multiplication techniques [9,10,11,12,13]. Additionally, efforts have been directed towards fine-tuning AC-to-DC and DC-to-DC conversion components to facilitate seamless energy transfer and precise voltage regulation [14,15,16,17,18]. To enhance the overall efficiency of these systems, energy storage solutions, such as supercapacitors and batteries, have been seamlessly integrated, enabling efficient energy storage and subsequent delivery [19,20,21,22,23]. Concurrently, researchers are actively engaged in developing matching circuits to ensure the smooth and uninterrupted transfer of RF energy from antennas to rectifiers [24,25,26,27,28]. On the front of power management strategies, the effective management of power plays a pivotal role in the success of RF harvesting systems, ensuring that harvested energy is efficiently utilized and directed to intended applications or networks. To achieve this, researchers have been pioneering adaptive algorithms and control systems designed to maximize power output, particularly when adapting to dynamic conditions [29,30,31,32,33]. In the realm of multi-band operations, there is ongoing exploration into intelligent switching and tuning mechanisms aimed at adapting to the ever-evolving RF environments and effectively extracting energy from diverse frequency bands [34,35,36,37,38]. Furthermore, research efforts are actively exploring strategies for energy storage management and distribution to optimize the efficient delivery of power to loads or networks [39,40,41,42].

Indeed, RF energy harvesting offers several compelling advantages, but it also comes with notable drawbacks [43]. One of the primary benefits is its sustainability and renewable nature, which helps lower carbon emissions [44]. However, the availability of ambient RF signals can be inconsistent, limiting the efficiency and reliability of this energy source [45]. Another significant advantage is that it eliminates the need for physical power connections and reduces battery replacements, benefiting devices in remote or hard-to-access areas [46]. Despite this, the power generated from RF energy harvesting is typically low, making it insufficient for high-energy-demand applications and restricting its usage to low-power devices [47].

RF energy harvesting can extend the operational lifespan of devices by providing a continuous power source, reducing maintenance and operational costs [48]. From a cost perspective, RF energy harvesting can reduce operational expenses by decreasing dependence on traditional power sources and minimizing battery replacement and maintenance costs [49]. However, the initial setup costs for RF energy harvesting systems can be high, requiring additional investment in infrastructure and equipment to maximize efficiency [50]. The mobility and flexibility offered by RF energy harvesting are significant, as devices powered by this technology are not constrained by wired power supplies [51].

On the other hand, RF energy harvesting has limited power output that may not suffice for devices with high energy demands, confining its application scope [52]. The technology’s effectiveness heavily depends on the availability and strength of ambient RF signals, which can be inconsistent. The design and integration of RF energy harvesting systems are complex, requiring advanced materials and technology, which can lead to higher initial costs and require specialized expertise [53]. Additionally, harvesting ambient RF energy can sometimes lead to interference with other RF-dependent systems, potentially affecting communication devices and networks [54]. Lastly, the conversion efficiency of RF energy harvesters is generally low, meaning a significant portion of the captured energy is lost during the conversion process, limiting the technology’s practical applications and effectiveness [55].

In this context, piezoelectric transformers, known for their compact size compared to traditional electromagnetic transformers, emerge as a key component, especially beneficial in portable and space-constrained applications [56]. Furthermore, these transformers demonstrate remarkable efficiency at specific resonant frequencies of the piezoelectric materials employed, underscoring their utility in the evolving landscape of wireless network technology [57]. These devices exploit the piezoelectric effect in specific materials, converting alternating voltage into mechanical vibrations and, subsequently, into electrical energy [58]. Their compact structure, high operational efficiency, and immunity to electromagnetic interference make them highly suitable for sensitive electronic applications [59]. However, it is essential to consider their limitations, such as a limited operating frequency range and lower power output compared to traditional electromagnetic transformers [60]. Notwithstanding these constraints, their application within wireless RF energy harvesting systems remains highly efficient, especially for low-energy requirements [61]. Specifically, piezoelectric transformers (PTs), resonating at specific frequencies, are integral to step-up oscillators in various roles. They can function as high-gain notch filters, especially when paired with JFETs, to enhance circuit efficiency by selectively amplifying certain signal frequencies [62,63]. Alternatively, PTs are utilized in the feedback loop of these oscillators to effectively amplify both positive and negative ultra-low voltages [64]. Additionally, they are employed in oscillator start-up converters, where they initiate oscillation at minimal initial voltage levels, enabling the oscillator to start effectively [65]. Beyond these applications, PTs are also instrumental in isolated DC–DC converters that convert waste heat energy into electricity. In this context, they not only improve efficiency but also reduce electromagnetic interference, contributing to a more efficient energy conversion process [66].

This study focuses on implementing RF energy harvesting using a piezoelectric transformer positioned directly after the antenna, a design not previously explored in existing research. Piezoelectric transformers enhance the efficiency of RF energy harvesting systems by converting mechanical energy into electrical energy, offering high power density, compact size, and reduced electromagnetic interference. However, integrating piezoelectric transformers can add complexity to the design and manufacturing process, potentially increasing costs and requiring specialized expertise for implementation.

## 2. Current Approach

The approach being discussed is depicted in Figure 1, which presents a block diagram of the proposed RF energy harvesting system. A key innovation in this methodology is the positioning of the piezoelectric transformer right after the front-end antenna. For this to be effective, the impedance of both the antenna output and the piezoelectric transformer (PT) input needs to be identical. In this design, they have both been configured to the standard value of 50 Ω.

The PT acts as a high-gain and efficient inverting amplifier. The captured RF signal is directly fed into the PT. The PT device is consists of input and output sections. The input section is horizontal employs the longitudinal piezoelectric effect; the corresponding displacement of the primary is mechanically coupled into the secondary, which causes the secondary to vibrate. The mechanical energy in the secondary is then converted to electrical energy, which is transferred to the secondary [67,68,69,70,71,72,73]. As illustrated in Figure 2, the input section is composed of PZT materials sandwiched between two metalizations, with the poling field direction along the thickness of the film. The poling field direction of the secondary section is vertical to the poling field of the primer section. The signal output from the piezoelectric transformer (PT) undergoes inversion and amplification with a high gain. Subsequently, a standard AC to DC conversion process is employed to transform the PT’s output signal into a usable DC bias.

### RF Power Harvesting Estimation with Piezoelectric Transformers

The power received from a transmitter in wireless power transfer systems is commonly determined by the Friis transmission equation. This equation is used to calculate the power received by an antenna from another antenna under line-of-sight conditions [74]:(1)$$P_{r}=\frac{P_{t}G_{t}G_{r}\lambda^{2}}{{\left(4\pi d\right)}^{2}}$$
where $P_{r}$, $P_{t}$, $G_{t}$, $G_{r}$, λ, and d are the power received, power transmitted, gain of the transmitting antenna, gain of the receiving antenna, wavelength of the transmitted signal, and the distance between the two antennas, respectively. The RF harvested power ($P_{o}$) produced by the PT can be expressed as [75]:(2)$$P_{o}=fs_{33}T^{2}V$$
where f is the operational frequency, V is the piezoelectric volume, $s_{33}$ is the elastic compliance, and T is the generated stress due to the input voltage. This stress is expressed as [76]:(3)$$T=-\frac{d_{31}c_{p}\upsilon }{t_{c}}$$
where $d_{31}$, $c_{p}$, $t_{c},$ and υ refer to the charge coefficient, elastic constant, thickness of the piezoelectric material, and the corresponding input voltage of the received power by the antenna, respectively. This voltage can be estimated using the antenna impedance ($z_{a}$) as follows [77]:(4)$$\upsilon =\sqrt{P_{r}z_{a}}$$

As is known, particularly in RF applications, a common impedance value is 50 Ω. Hence, the harvested power can be expressed as:(5)$$P_{o}=\left(\frac{lw}{t_{c}}\right)\left(s_{33}d_{31}^{2}c_{p}^{2}\right)\left(\lambda P_{t}\right)\left(G_{t}\right)\left(\frac{c/\sqrt{\varepsilon_{r}}}{{{16\pi }^{2}d}^{2}}\right)\left(G_{r}z_{a}\right)$$
where l and w are the length and width of piezoelectric materials, respectively. c and $\varepsilon_{r}$ represent the speed of light in a vacuum and the dielectric constant of the transmitted media, respectively. In Equation (5), the initial term describes the physical properties of the piezoelectric material. The second term accounts for the mechanical and piezoelectric parameters. The third term signifies the wavelength and power of the transmitted signal, and the fourth term pertains to the transmitting antenna. The fifth term corresponds to the properties of the medium through which the signal wave propagates. Lastly, the sixth term is associated with the receiving antenna.

## 3. Experimental Setup

The employed setup is depicted in Figure 3a. A function generator was used to feed the transmitted antenna (Tx). Two identical small active loop antennas (GA800 Radio Shortwave, WiMo Antennen und Elektronik GmbH, Landau/Pfalz, Germany), which radiate from 10 kHz up to the VHF band, were used for transmission and reception. The antennas had an input impedance of 50 Ω and a gain of 20 dB, with a loop size of 26 cm and an operating temperature range from −65 to 75 °C [78]. The two antennas were placed far from each other to enable far-field radiation. The receiving antenna (Rx) was directly connected to the piezoelectric transformer (PT). The piezoelectric transformer used is shown in Figure 3b; it had one input and one output. The transformer was composed of lead zirconate titanate (PZT), with a working frequency of 50 kHz and dimension of 35 × 6 × 1.2 mm^3^ [79]. Figure 3b illustrates that the piezoelectric transformer (PT) features three terminals: two on the input side, marked as negative and positive, and one on the output side. The output side of the device was linked to an AC to DC converter circuit, as depicted in Figure 3c. Additionally, the negative terminals of both the AC/DC converter and the piezoelectric transformer (PT) were interconnected. The next step was to convert the AC output voltage signal from the PT to DC. The AC to DC converter circuit is depicted in Figure 3c. The full-wave rectifier circuit consisted of four germanium diodes [80] and a smoothing capacitor of 1000 µF. The output of the AC/DC circuit was connected to an oscilloscope to display the output DC voltage. Due to their low forward voltage and superior power handling capabilities, germanium diodes with the model 1N34A (Jameco Electronics, Belmont, CA, USA) and a forward bias of 0.3 V were utilized. This type of diode is known for its low leakage current and robust mechanical strength. Each diode had a voltage drop of 0.3 V in a bridge configuration; therefore, the total voltage across the bridge was 0.6 V.

## 4. Measurements and Analysis

Prior to conducting the measurements for the suggested circuit, the frequency responses of the antenna and the PT transformer were measured and analyzed individually. These responses are illustrated in Figure 4a,b for the antenna and the PZT transformer, respectively. The antenna was measured using a Rohde and Schwartz R&S^®^ZVL network analyzer (Rohde & Schwarz Emirates L.L.C., Abu Dhabi, United Arab Emirates) [81]. The data presented in Figure 4a illustrate the return loss in dB of the antenna versus frequency. It demonstrated a resonance band approximately at 50 kHz ± 10 kHz, making it highly suitable for this demonstration. The frequency characteristics of the PZT transformer were measured using a Gamry Reference 3000 analyzer (Gamry Instruments, Warminster, PA, USA) [82]. Figure 4b depicts the measured input impedance magnitude with frequency. The PT exhibited a series resonance at 50 kHz ± 5 kHz and parallel resonance at 57 kHz.

The system depicted in Figure 1 was realized, as demonstrated in Figure 3. Figure 5 presents the output voltage at the antenna with and without the use of a piezoelectric transformer for a transmitted sinusoidal voltage with frequency of 50 kHz. The PT acted as an amplifier, as the sinusoidal signal was amplified 12 times compared to the input signal, inverting the signal. The black signal represents the measurement taken immediately after the antenna, in the absence of PT, showing only noise and tending towards zero amplitude. In contrast, the red signal is the measurement taken after the PT connected to the antenna. Without the transformer, the antenna’s output approached zero, whereas with the transformer, it reached a peak of 650 mV. This represents a significant achievement in this research. It is important to note that both the piezoelectric transformer (PT) and the antenna had an identical input impedance of 50 Ω.

The antennas were positioned 2 m apart to ensure far-field conditions. The transmitting antenna was supplied with a sinusoidal signal of 50 kHz and a voltage of 10 V peak-to-peak. The choice of frequency was dictated by the commercial availability of the piezoelectric transformer (PT). While PTs can be adapted to function at higher frequencies, this requires significant technological advancements. Any mismatch in this impedance would result in a corresponding attenuation of the output signal, as will be demonstrated in Section 5.

Figure 6 plots the measured rise of DC voltage across the capacitor. The capacitor used in this setup is an aluminum electrolytic type with a capacitance of 1000 µF and a voltage rating of 25 V. This specification of capacitor is commonly employed. Figure 5 shows the accumulation of DC voltage in the capacitor over time, with the time measured in minutes, and presents a detailed view of the charging process. In the beginning phase, from 0 to 3 min, there was a slow and gradual increase in voltage. A significant shift occurred between 3 and 4 min, during which the voltage saw a sharp increase, leaping from 0.44 V to 1.32 V. Then, the voltage showed a consistent and linear rise. This steady growth suggests a stable and continuous rate of charge accumulation. Overall, the charging pattern of the capacitor began with a gradual buildup, followed by a rapid spike, and then transition into a phase of steady and linear growth, with a slight fluctuation observed towards the end. This pattern provides valuable insights into the capacitor’s charging dynamics and the efficiency of the system over an extended period.

It was also important to confirm that a bridge rectifier circuit was used. Conventionally, when the antenna is directly connected to the rectifier circuit, the rectifier’s output should not exceed the RMS value of the input signal from the antenna. However, in our innovative design using a piezoelectric transformer (PT), the voltage delivered by the antenna was further amplified, as the PT functioned as an inverting amplifier with high gain at the resonance frequency. In both cases, the rectifier’s output should not surpass the input RMS value.

Next, the focus was on the power harvested and delivered to the load. Figure 7 illustrates the measured power delivered to different loads at a resonance frequency of 50 kHz. The load power was evaluated before and after the AC to DC converter. In both cases, the optimal load corresponding to maximum power was recorded at 100 kΩ. At resonance, the piezoelectric transformer (PT) behaved resistively, and the optimal load matched the internal resistance of the harvester. With the use of the AC to DC converter, the delivered power improved due to the loading effect shown in Figure 8.

The loading effect was applied, and the results are shown in Figure 8. This figure displays the magnitude of input impedance versus frequency under various scenarios: antenna alone, antenna with the rectifier circuit, antenna with the piezoelectric transformer (PT), and antenna with both the PT and rectifier circuit. The integration of the rectifier circuit with the antenna showed a negligible effect. As demonstrated in Figure 8, the inclusion of the piezoelectric transformer influenced the input impedance performance. Despite these modifications, the input impedance indicates that the system comprising the antenna, PT, and rectifier circuit continues to resonate at 50 kHz, maintaining effective performance.

## 5. Piezoelectric Transformer Effect on Rf Energy Harvester Performance

This section examines how the properties of the piezoelectric transformer influence the performance of the RF energy harvester. Additionally, it provides an analysis of RF performance in relation to distance. The properties of the piezoelectric transformer, including its resonant frequency, impedance matching, material composition, and structural design, critically influence the performance of the RF energy harvester. Additionally, the distance between the transmitting and receiving antennas significantly impacts RF performance, with greater distances leading to reduced harvested power. These findings underscore the importance of optimizing both the PT properties and the system setup to enhance the efficiency of RF energy harvesting systems.

Figure 9 presents the output voltage characteristics of the piezoelectric transformer in relation to frequency, input voltage, and input impedance mismatch. Figure 9a illustrates the output voltage as a function of input frequency, showing a peak at the resonance frequency. This resonance frequency depends on dimensions, thicknesses, and involved piezoelectric materials constants [83,84]. Figure 9b displays the voltage transfer characteristics, where the amplitude of the input AC signal was varied from zero to 1 volt at the resonance frequency. The effective electromechanical coupling coefficient, piezoelectric voltage constants, and the optimal size of a piezoelectric transformer all contribute to the output voltage [85]. As revealed, the performance of the piezoelectric transformer showed a linear gain up to 0.8 volts, after which it began to exhibit nonlinear behavior at higher voltages. The piezoelectric transformer demonstrated a gain of 18 Volts/Volts in the lower range, as shown in Figure 9b. This gain could be further enhanced by using a proper piezoelectric transformer with a multilayer design [86]. Figure 9c summarizes the impedance mismatch at the input section for an input voltage of 0.6 volts with an AC signal at the resonance frequency. Clearly, the mismatch would degrade the output voltage and consequently reduce the output power. Therefore, the design rules for both the antenna and the piezoelectric transformer should ensure a perfect match. If this is not feasible in rare applications, then a matching circuit should be used [87].

Figure 10 illustrates the relationship between output power and the distance between the two antennas. The far-field radiation distance was calculated to be 20 μm. As the distance increased, the power decreased inversely with the square root of the distance. These measurements were conducted at the optimal load and resonance frequency. Naturally, the frequency characteristics, dimensions, and topology of the antenna significantly influence its power performance. The utilized antenna, an active loop antenna (GA800 Radio Shortwave) radiating from 10 kHz up to the VHF band, was picked to demonstrate the concept of the current work and was not optimized to achieve maximum radiation and directivity at the 50 KHz resonance frequency. The power and efficiency of the system could be further enhanced by using an antenna designed for higher gain and low loss.

Furthermore, it is worth adding that there are no constraints on the antenna size; the only requirement is that the resonance frequency of both the antenna and the piezoelectric transformer must be identical. The loop antenna used here had a wide resonance frequency range, including 50 KHz. The size of the antenna may vary depending on the application. We believe that antenna design rules can be adopted to create more compact antennas. A loop antenna was chosen for this study due to its commercial availability, but other types or designs can also be used.

## 6. Benchmarking with State of Art

When compared with the state of the art, none have been reported using a piezoelectric transformer. Adami et al. introduced a 2.45-GHz flexible wireless power harvesting wristband capable of generating a net DC output from an extremely low RF input of −24.3 dBm, setting a new low in system sensitivity for such devices [88]. It features a novel all-textile antenna with over 62% radiation efficiency, a highly efficient rectifier on a rigid substrate, and an innovative contactless electrical connection. The wristband, incorporating a self-powered boost converter, achieves an exceptional end-to-end efficiency of 28.7% at −7 dBm, significantly surpassing the state of the art in energy harvesting. Assimonis et al. proposed an innovative RF energy harvesting supply with high efficiency and sensitivity [89]. Utilizing a single-series circuit with a double diode on a cost-effective, though typically less efficient, FR-4 substrate, the design achieves a rectenna efficiency of 28.4% at 20-dBm input. The system’s sensitivity is enhanced through series-configured rectennas, enabling operation at very low RF power densities. Additionally, the incorporation of a self-starting commercial boost converter further optimizes performance, demonstrating the system’s capability to effectively power a scatter radio sensor under varying power densities. Mattsson et al. designed a novel dual-band rectenna for RF energy harvesting, featuring a unique design with two concentric square patches for operation in the 2.4 and 5.5 GHz Wi-Fi bands [90]. The antenna showcases gains of 7.52 and 7.26 dBi at these frequencies. It employs a full-wave Greinacher voltage doubler rectifier for each polarization, effectively quadrupling the output voltage. This design enables the rectenna to act as a direct power source for electronics that require a differential source, demonstrating a significant advancement in RF energy harvesting technology. Shen et al. present a novel dual-frequency microstrip rectifier for wireless power transmission systems, operating at 1.8 GHz and 2.4 GHz. It features a unique cross-shaped matching stub to handle different impedances at these frequencies [91]. Additionally, a DC-pass filter effectively blocks unwanted frequencies. The rectifier achieves impressive RF–DC conversion efficiencies of 77.5% and 75.1% at 1.8 GHz and 2.4 GHz respectively with a 10 mW input, marking a significant advancement in dual-band RF energy harvesting. Fan et al. have built a high-efficiency rectifier-booster regulator (RBR) with an impedance matching converter for WLAN energy harvesting at 2.45 GHz [92]. It features a novel RBR design, combining a Greinacher rectifier with Cockcroft–Walton charge pumps for effective RF to DC conversion and voltage boosting.

The system employs a unique flower-shaped, dual-polarized antenna and a boost converter for optimal energy harvesting, storing energy in a supercapacitor. Experimentally, the RBR shows a 1.7-V output at −10 dBm input power and achieves a 3.4 voltage boost with an 85% voltage conversion rate, proving its potential for IoT applications. Pham et al. showed an antenna capable of operating across three frequency bands, including cellular network frequencies (900 MHz and 1900 MHz) and Wi-Fi (2.4 GHz) [93]. Utilizing a blend of design techniques, including composite right/left-hand transmission lines, they also proposed a highly efficient, sensitive triple-band rectifier. This rectifier achieves maximum conversion efficiencies of 80%, 46%, and 42% at specific frequencies. Remarkably, the triple-band RF energy harvesting system can collect significantly more power compared to single-band systems, demonstrating its enhanced capability in ambient energy harvesting. Table 1 in the document benchmarks this study against others in the field by comparing key parameters like load resistance, output voltage, input power, and efficiency. This study notably showcases a significant improvement, achieving an 88% efficiency with a high load resistance of 40 kΩ, and an output voltage of 0.65 V at 9.7 dBm input power.

Moreover, in the context of GHz frequency wireless energy harvesting, GHz Bulk Acoustic Wave (BAW) Piezoelectric Transformers represent a viable solution [94]. These devices leverage a synergistic blend of thin films characterized by high and low dielectric constants, thereby facilitating enhanced voltage gain. Additionally, an alternative approach involves miniaturizing the dimensions of the existing piezoelectric transformer model. Indeed, the presented approach could be considered a scalable RF energy harvesting system. Piñuela et al. have concentrated on creating harvesters suitable for urban and semi-urban environments, focusing on ambient RF power levels. Their designs achieved efficiencies of up to 40%, with about half of the tested stations suitable for energy harvesting [95]. This concept has the potential to be expanded into multiband arrays for broader applications, offering a competitive edge.

It is worth adding that by integrating the current approach into the mobile WPIoT system described in [96], the overall efficiency of RF energy harvesting can be enhanced. This will improve the network’s sustainability by reducing dependency on external energy sources and extending the operational lifespan of mobile IoT nodes. Moreover, incorporating energy harvesting feedback into the resource allocation model can further optimize task distribution, helping to achieve the primary objective of minimizing the Age of Information.

By employing multiple antenna elements, each tuned to different frequency bands and connected to a matching piezoelectric transformer (PT), the harvested energy can be amalgamated using a power combiner. This combined energy could then power more demanding electronic devices or sensors. Additionally, incorporating tunability into the piezoelectric transformer can adapt to the unpredictability of various RF bands. The RF harvester, as designed, is capable of gathering energy from numerous sub-frequency bands, thereby boosting the output voltage ratios in proportion to the number of RF bands. Enhancing efficiency is possible through the development of innovative system architectures for RF harvesters [37]. This approach is beneficial for remote or inaccessible locations, for devices with limited power needs, or in situations where ambient RF energy (such as from communication networks or industrial equipment) is available but varies over time and space. The ability to scale this system up or down makes it a promising solution for a sustainable and autonomous energy supply for a wide range of electronic devices.

## 7. Conclusions

This study presents an innovative RF energy harvesting system that utilizes a piezoelectric transformer (PT) as a high-gain inverting amplifier to efficiently convert ambient RF signals into electrical energy, offering a sustainable power source. The PT, composed of lead zirconate titanate (PZT), was positioned immediately after the receiving loop antenna, with both components impedance-matched at 50 Ω. The output of the PT was connected to an AC-to-DC converter circuit to generate usable DC voltage. The experiments demonstrated that the PT significantly amplifies the RF signal, achieving a peak output of 650 millivolts compared to negligible output without the transformer. The system reached maximum harvested power at the PT’s resonance frequency of 50 kHz, with an optimal load of 40 kΩ. Input impedance measurements confirmed that the system, consisting of the antenna, PT, and rectifier circuit, maintains stable resonance at 50 kHz, ensuring effective performance. The performance of the system was also influenced by the PT properties and the distance between antennas, with greater distances reducing harvested power. Frequency response measurements confirmed the PT’s linear gain up to certain input voltages, underscoring its efficiency. The overall system efficiency was characterized at 88%, highlighting the potential of PTs for high-efficiency RF energy harvesting. These findings lay a solid foundation for future improvements in antenna design and advanced PT configurations, enabling more efficient RF energy harvesting systems. This is especially crucial for applications requiring a remote power supply, where energy efficiency and signal amplification are essential.

## Figures and Tables

**Figure 1 sensors-24-07111-f001:**
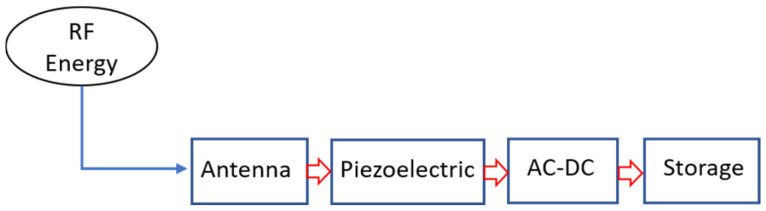
Proposed RF energy harvesting incorporating the piezoelectric transformer immediately after the antenna.

**Figure 2 sensors-24-07111-f002:**
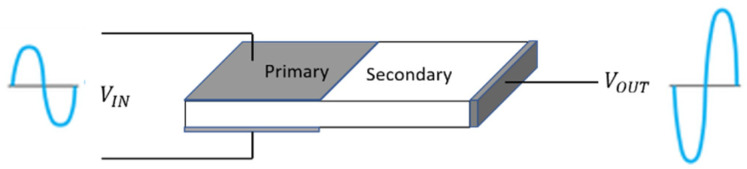
Layout of the Rosen-type piezoelectric transformer consists of a primary and a secondary section that are mechanically coupled by the transversal area of the PT. The PT is acting as voltage-voltage inverted amplifier.

**Figure 3 sensors-24-07111-f003:**
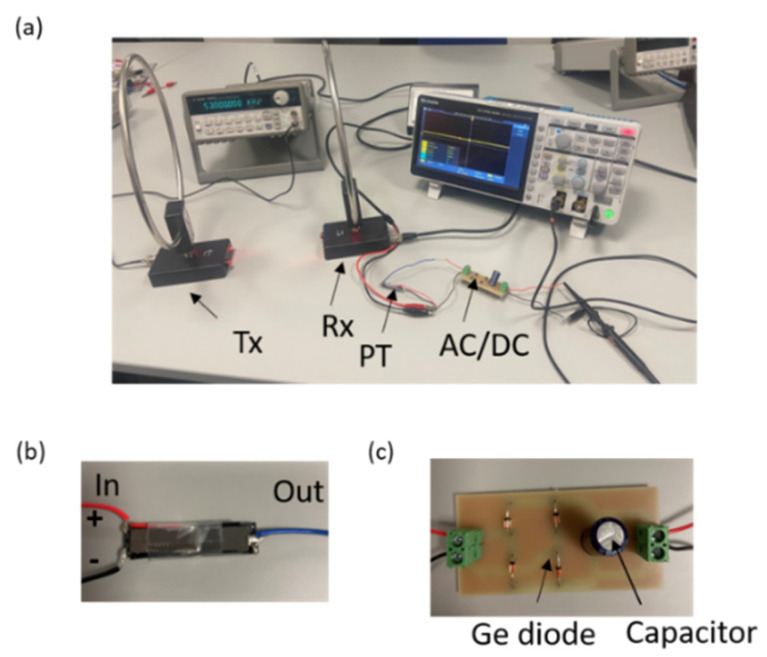
Experimental configuration: (**a**) detailed view of the setup, (**b**) piezoelectric transformer (PT) with dimensions of 35 × 6 × 1.2 mm^3^, and (**c**) AC to DC converter with a smoothing capacitor capable of storing DC charges.

**Figure 4 sensors-24-07111-f004:**
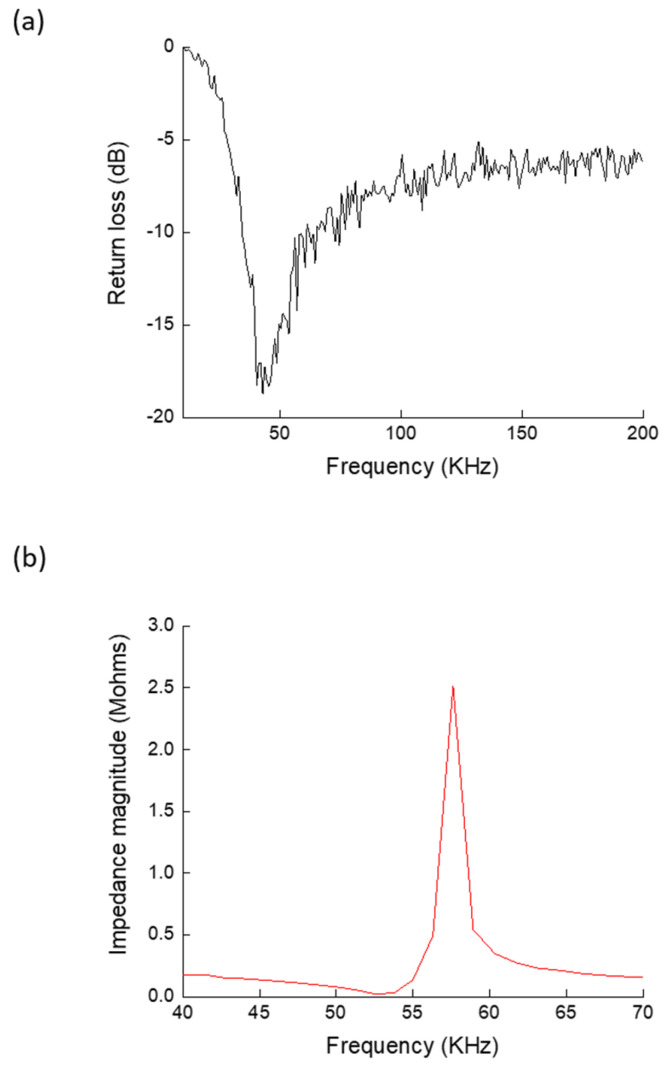
Measured frequency characteristics for (**a**) the antenna, and (**b**) the PT transformer, showing the required matching in resonance frequency.

**Figure 5 sensors-24-07111-f005:**
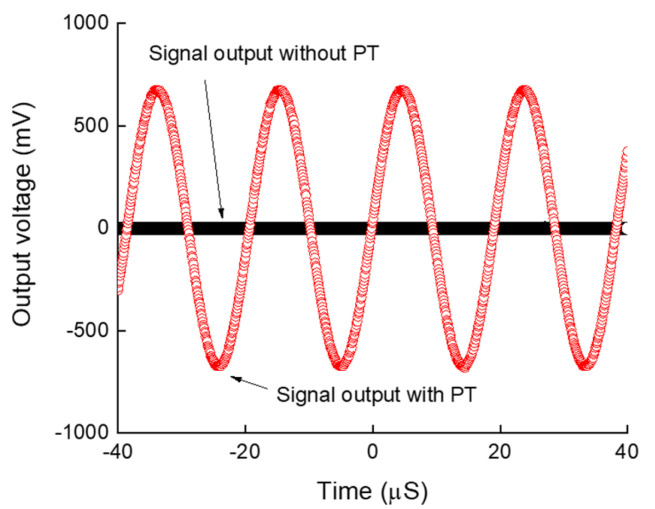
Measured signals after the front-end antenna with and without the use of a piezoelectric transformer, demonstrating the novelty and innovation of this study.

**Figure 6 sensors-24-07111-f006:**
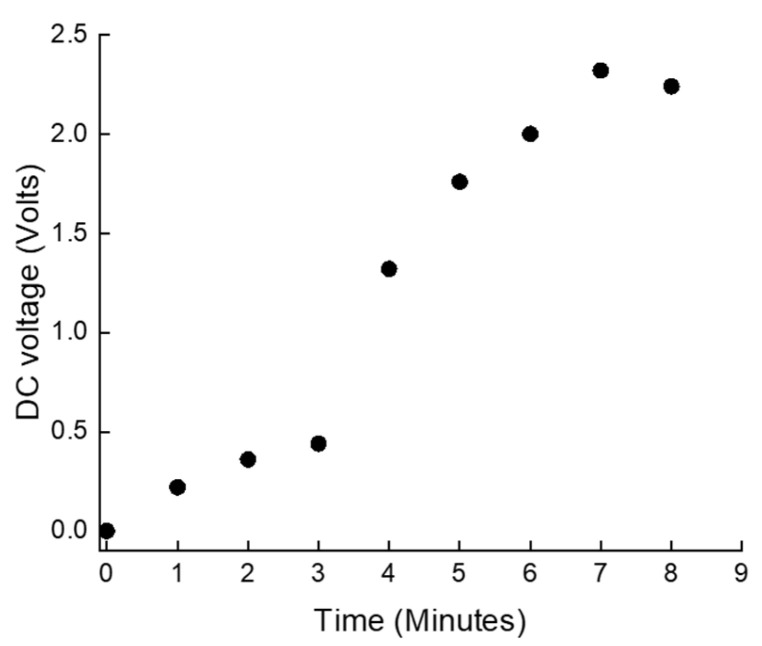
Rise of DC voltage of the capacitor. The rise of DC voltage across the capacitor refers to the gradual increase in the direct current (DC) voltage level stored on the capacitor’s plates. This occurs as the capacitor accumulates electric charge over time when connected to a DC power source.

**Figure 7 sensors-24-07111-f007:**
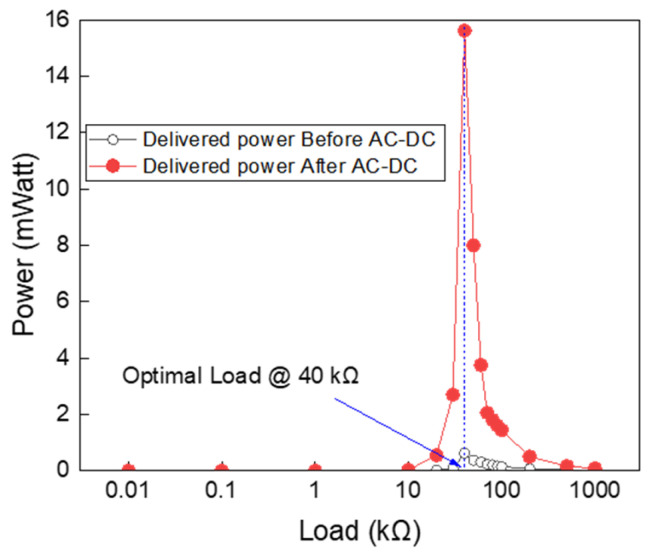
Measured power versus load before and after the AC to DC converter, demonstrating the functionality of the AC to DC converter.

**Figure 8 sensors-24-07111-f008:**
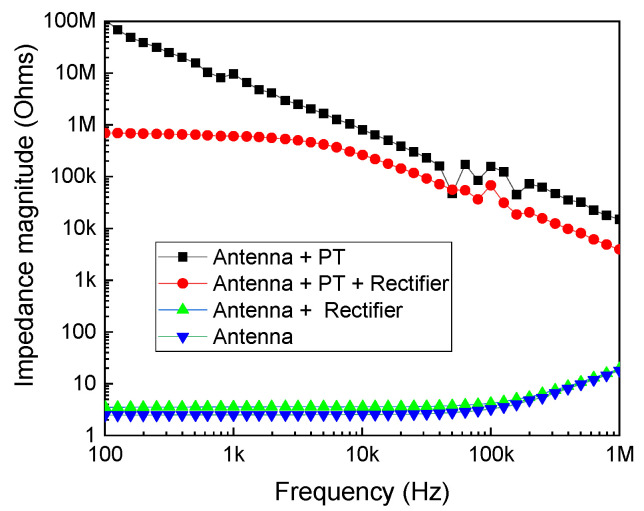
Input impedance measurements to study the loading effect.

**Figure 9 sensors-24-07111-f009:**
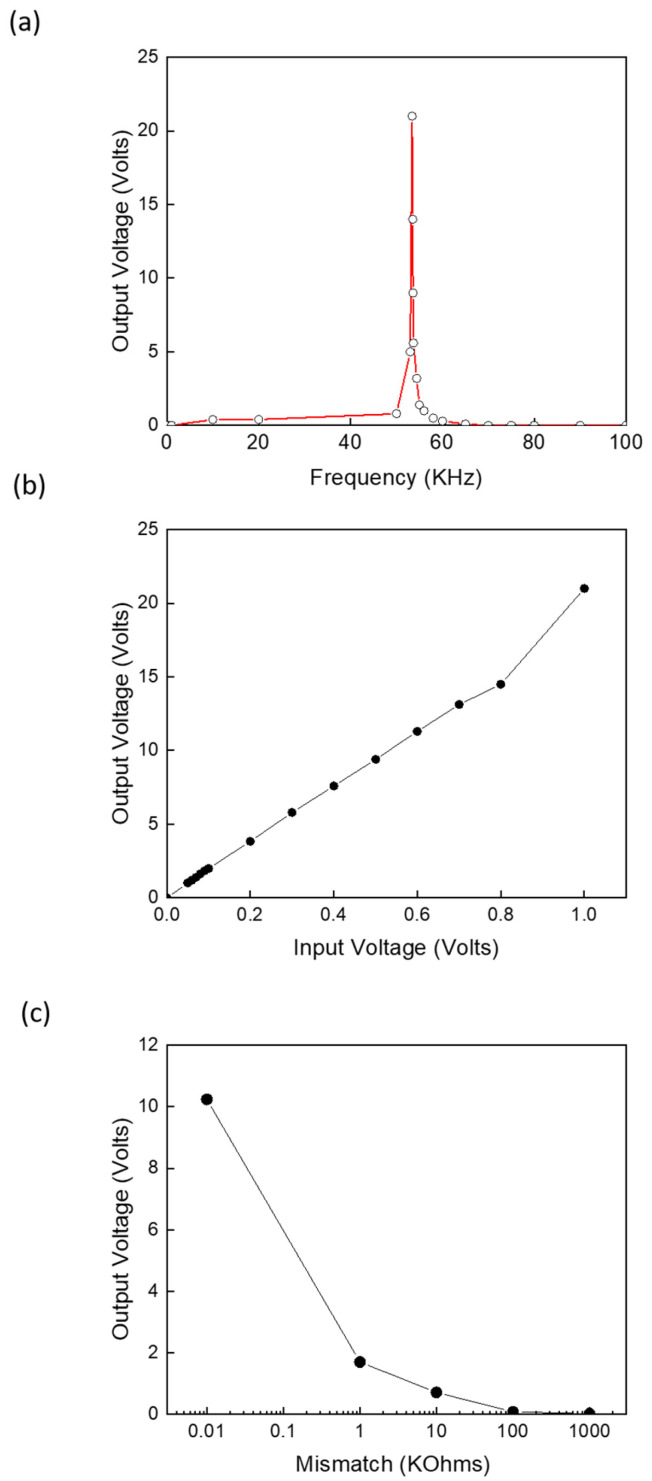
Piezoelectric transformer output voltage measurements: (**a**) voltage versus frequency at an input voltage of 600 mVolts, (**b**) voltage versus input voltage at resonance frequency, and (**c**) voltage versus impedance mismatch at an input voltage of 600 mVolts and resonance frequency.

**Figure 10 sensors-24-07111-f010:**
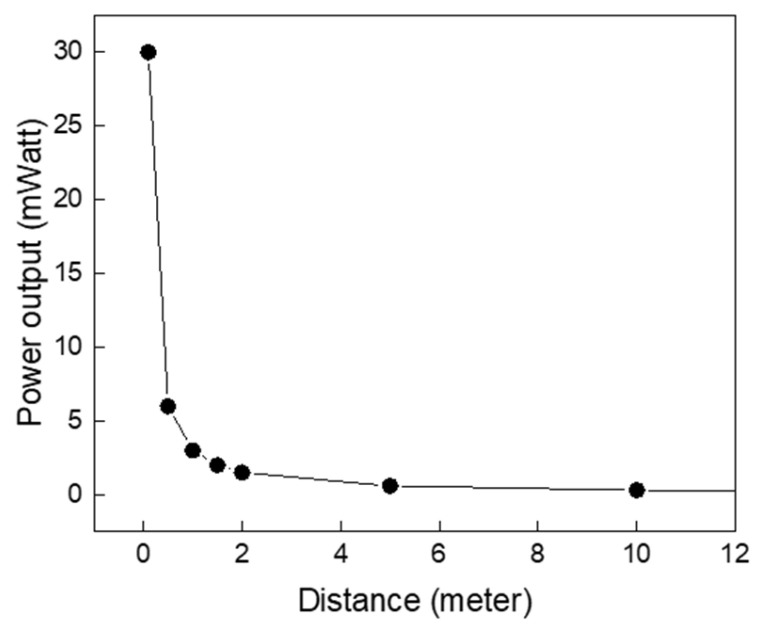
Output power versus separation distance between the two antennas at the optimum load and resonance frequency.

**Table 1 sensors-24-07111-t001:** Comparison between different reported works.

Ref.	Load Resistance	Output Voltage	Input Power (dBm)	Efficiency
[70]	3 kΩ	0.41 V	−10	56.00%
[71]	9.53 kΩ	0.65 V	−10	44.20%
[72]	12 kΩ	0.49 V	5	36.00%
[73]	917 Ω	0.5 V	12	75.10%
[74]	12 kΩ	0.24 V	16	43.00%
[75]	20 kΩ	1.7 V	13	37.50%
This Work	40 kΩ	0.65 V	9.7	88%

## Data Availability

Dataset available on request from the authors.

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
