# Peer review of "Enhanced RF Energy Harvesting System Utilizing Piezoelectric Transformer"

_sensors, 2024, doi:10.3390/s24227111_

Round 1

Reviewer 1 Report

Comments and Suggestions for Authors

This study highlights the effectiveness of using a piezoelectric transformer for harvesting RF energy from surrounding areas. This paper is easy to follow. Some issues should be addressed as follows.

1.     The technical depth and main contributions in abstract should be better summarized and emphasized.

2.     Based on the first paragraph of the introduction, it is obvious that the background of this paper includes wireless communication and RF energy harvesting. Recent high quality works should be introduced, such as Distributed DDPG-based resource allocation for age of information minimization in mobile wireless-powered Internet of Things, IEEE IoTJ.

3.     There are so many references in this paper, while few of them are introduced with detailed contributions in the introduction.

4.     Many references, such as [30]-[34], are outdated and should be updated with recent high quality works on wireless communications and RF EH.

5.     In table 1, some comparison systems should be updated by recent works to evaluate the efficiency performance.

6.     Impacts of more vital network parameters on network performance should be provided. Besides, more detailed observations and reasons behind the observation in experiments should be provided.

7.     Important results and findings should be better summarized in conclusion.

Comments on the Quality of English Language

Moderate editing of English language is required for this manuscript.

Author Response

Response to the Comments and Suggestions from Reviewer 1:

This study highlights the effectiveness of using a piezoelectric transformer for harvesting RF energy from surrounding areas. This paper is easy to follow. Some issues should be addressed as follows.

Thank you very much for your thorough and positive review of our paper. We truly appreciate your valuable time and feedback.

  1. The technical depth and main contributions in abstract should be better summarized and emphasized. Authors response: Please find the revised abstract below.

“RF energy harvesting converts ambient signals into electrical power, providing a sustainable energy source. This study demonstrates the use of a piezoelectric transformer for efficient RF energy harvesting. In this work, a piezoelectric transformer (PT) is employed as a high-gain, efficient inverting amplifier to enhance RF wireless energy harvesting. The PT, composed of lead zirconate titanate (PZT), is placed after the receiving loop antenna, with its output connected to an AC-to-DC converter circuit. Maximum harvested power was observed at the PT's resonance frequency of 50 kHz, with an optimal load of 100 kΩ. The system, comprising the antenna, transformer, and rectifier circuit, continues to resonate at 50 kHz, as confirmed by input impedance measurements, demonstrating stable and effective performance. The overall system efficiency was characterized to be 88%.”

  1. Based on the first paragraph of the introduction, it is obvious that the background of this paper includes wireless communication and RF energy harvesting. Recent high quality works should be introduced, such as Distributed DDPG-based resource allocation for age of information minimization in mobile wireless-powered Internet of Things, IEEE IoTJ. Authors response: We have included the following text in the benchmarking section to emphasize the potential application of this approach in Wireless-Powered IoT, citing the recent advancements by Zheng et al. as follows:

“By integrating the current approach into the mobile WPIoT system described in [96], the overall efficiency of RF energy harvesting can be enhanced. This will improve the network's sustainability by reducing dependency on external energy sources and extending the operational lifespan of mobile IoT nodes. Moreover, incorporating energy harvesting feedback into the resource allocation model can further optimize task distribution, helping achieve the primary objective of minimizing the Age of Information.”

[96] K. Zheng, R. Luo, X. Liu, J. Qiu and J. Liu, "Distributed DDPG-Based Resource Allocation for Age of Information Minimization in Mobile Wireless-Powered Internet of Things," in IEEE Internet of Things Journal, vol. 11, no. 17, pp. 29102-29115, 2024. doi: 10.1109/JIOT.2024.3406044.

  1. There are so many references in this paper, while few of them are introduced with detailed contributions in the introduction. Authors response: The paper includes a total of 96 references, with 66 cited in the introduction section to underscore the significance of the topic and provide a concise overview of the state of the art. An additional 11 references are used in the benchmarking section to compare current results with previous work, and to highlight the potential for scaling the proposed approach to other frequency ranges and systems. The remaining references are utilized in the modeling section. All cited references are relevant to the paper’s topic. Given that this is a novel development, the extensive citations serve to demonstrate the authors' comprehensive awareness of the existing literature and to establish that the work presented has not been previously reported.

  1. Many references, such as [30]-[34], are outdated and should be updated with recent high quality works on wireless communications and RF EH. Authors response: The references cited are of high quality, predominantly sourced from IEEE journals with high impact factors. The authors believe it is not advisable at this stage to replace any of the references. However, they are open to adding more references based on the reviewer's suggestions. The authors have already incorporated the reference suggested by the respected reviewer earlier in point 2, particularly from Liu's group. Therefore, the authors kindly request the reviewer to reconsider this comment.

  1. In table 1, some comparison systems should be updated by recent works to evaluate the efficiency performance. Authors response: The authors believe that Table 1 is well-organized and accurately reflects the efficiency of the reported systems. It is important to note that recent work does not necessarily indicate higher efficiency. In fact, a recent comprehensive review authored by six IEEE fellows (link provided below) indicates that the maximum efficiency reported in recent studies is around 37%. Therefore, while we acknowledge the importance of incorporating recent developments, the assumption that more recent works offer higher efficiency is not always the case. We kindly request the reviewer to consider this context. However, we remain open to updating the table if there are any specific recent works that the reviewer recommends for inclusion.  

https://ieeexplore.ieee.org/stamp/stamp.jsp?tp=&arnumber=10091717

  1. Impacts of more vital network parameters on network performance should be provided. Besides, more detailed observations and reasons behind the observation in experiments should be provided. Authors response: This work is currently at the hardware development and proof-of-concept stage and has not yet been integrated with any wireless networks, such as IoT, to evaluate its impact on key network performance parameters. That said, the authors plan to contact the research groups of Professor J. Liu and Professor K. Zheng to explore potential collaborations for integrating the proposed solution with their outstanding work.

  1. Important results and findings should be better summarized in conclusion. Authors response: Please find the revised conclusion below.

“This study presents an innovative RF energy harvesting system that utilizes a piezoelectric transformer (PT) as a high-gain inverting amplifier to efficiently convert ambient RF signals into electrical energy, offering a sustainable power source. The PT, composed of lead zirconate titanate (PZT), is positioned immediately after the receiving loop antenna, with both components’ impedance-matched at 50 Ω. The output of the PT is connected to an AC-to-DC converter circuit to generate usable DC voltage. Experiments demonstrated that the PT significantly amplifies the RF signal, achieving a peak output of 650 millivolts compared to negligible output without the transformer. The system reached maximum harvested power at the PT's resonance frequency of 50 kHz, with an optimal load of 100 kΩ. Input impedance measurements confirmed that the system, consisting of the antenna, PT, and rectifier circuit, maintains stable resonance at 50 kHz, ensuring effective performance. The performance of the system is also influenced by the PT properties and the distance between antennas, with greater distances reducing harvested power. Frequency response measurements confirmed the PT's linear gain up to certain input voltages, underscoring its efficiency. The overall system efficiency was characterized at 88%, highlighting the potential of PTs for high-efficiency RF energy harvesting. These findings lay a solid foundation for future improvements in antenna design and advanced PT configurations, enabling more efficient RF energy harvesting systems. This is especially crucial for applications requiring remote power supply, where energy efficiency and signal amplification are essential.”

Reviewer 2 Report

Comments and Suggestions for Authors

The authors presented a study combining RF technology with piezoelectricity for energy harvesting. They conducted a thorough investigation with experimental studies. However, there are some critical points that need to be addressed:

  1. There is confusion regarding the impedance match of 50 Ohms mentioned in the text, while Figure 7 shows a piezoelectric load of 100kOhms. What accounts for this difference? Can the impedance of RF make this equivalent despite the significant disparity?

  2. In Figure 7, there is a large gap between the steps of the piezoelectric load, ranging from 10kOhms to 100kOhms. Typically, the optimal piezoelectric resistance load lies around 20kOhms to 50kOhms, making the reported load of 100kOhms incorrect.

  3. The dimensions of the piezoelectric generator should be provided in Figure 3b.

  4. The operating resonant frequency of the piezoelectric samples is stated to be 50 kHz, which is quite high. However, the reference given (Ref.79) indicates a resonant working frequency of 67kHz. The authors need to present the dimensions of this generator and conduct a finite element or analytical investigation to calculate the correct resonant frequency of the system they tested.

  5. The output power for piezoelectric materials is not as simple as Equation 2 (Ref.75) suggests. Ref.75, from the year 2002, used very simple equations to estimate piezoelectric power, which are now known to be incorrect. The authors need to re-establish the analytical equations.

Comments on the Quality of English Language

The English needs to be corrected. 

There are mistakes such as "Line 33 Components them" and "Line 49 realm of ...".

Author Response

Response to the Comments and Suggestions from Reviewer 2:

The authors presented a study combining RF technology with piezoelectricity for energy harvesting. They conducted a thorough investigation with experimental studies. However, there are some critical points that need to be addressed:

Thank you very much for your thorough and positive review of our paper. We truly appreciate your valuable time and feedback.

  1. There is confusion regarding the impedance match of 50 Ohms mentioned in the text, while Figure 7 shows a piezoelectric load of 100kOhms. What accounts for this difference? Can the impedance of RF make this equivalent despite the significant disparity? Authors response: The RF antenna has an impedance of 50 ohms. The PT (presumably a power transfer or processing component) has both input and output impedances. To minimize signal distortion, the input impedance of the PT should match the RF antenna's impedance or be in the same order of magnitude. Similarly, the output port of the PT is connected to the load, which also needs to be of a similar impedance to avoid further signal distortion. This is the reason for considering an optimal load (the optimal load matches the internal resistance of the whole harvester system). To further enhance energy conversion, it would be beneficial to insert a matching circuit between the antenna and the PT. This matching circuit will help optimize the impedance matching, reducing signal reflection and ensuring more efficient power transfer. This is what is meant by impedance matching to 50 ohms: the matching circuit ensures that the impedance of the antenna, PT, and load are all aligned at 50 ohms, minimizing signal reflections and maximizing power transfer efficiency.

  1. In Figure 7, there is a large gap between the steps of the piezoelectric load, ranging from 10kOhms to 100kOhms. Typically, the optimal piezoelectric resistance load lies around 20kOhms to 50kOhms, making the reported load of 100kOhms incorrect. Authors response: We agree with this comment, and due to the gap, the load was initially recorded at 100K Ohms. We have now remeasured at 20K, 30K, 40K, 50K, 60K, 70K, 80K, and 90K Ohms, and have updated Figure 7 accordingly. The optimum load is now measured at 40K Ohms. The revised figure is attached here

FIGURE 7. Measured power versus load before and after the AC to DC converter demonstrating the functionality of AC to DC converter.

  1. The dimensions of the piezoelectric generator should be provided in Figure 3b. Authors response: The dimensions have been provided in the caption of Figure 3b as requested (35 × 6 × 1.2 mm3). Additionally, they have already been mentioned in the EXPERIMENTAL SETUP section, line 13 in the original submission.

  1. The operating resonant frequency of the piezoelectric samples is stated to be 50 kHz, which is quite high. However, the reference given (Ref.79) indicates a resonant working frequency of 67kHz. The authors need to present the dimensions of this generator and conduct a finite element or analytical investigation to calculate the correct resonant frequency of the system they tested. Authors response: We believe there may be some misunderstanding. To clarify, we are not harvesting energy from vibration, where the frequency would be low, around 100 Hz. Instead, we are harvesting RF ambient energy, which operates at much higher frequencies. The core and focus of this work is the use of an energy transformer functioning as an inverted amplifier, not as a piezoelectric generator. We kindly ask the respected reviewer to take this distinction into consideration. Additionally, piezoelectric transformers are commercially available and well characterized, as per the following link:
    https://www.steminc.com/PZT/en/single-layer-piezo-electric-transformer-50-khz.

Regarding reference 79, the link has been updated, and we apologize for the mistake for mentioning 67KHz. The reference has been updated. We have already characterized the device using impedance and voltage versus frequency measurements, as reported in Figures 4b and 9a.

  1. The output power for piezoelectric materials is not as simple as Equation 2 (Ref.75) suggests. Ref.75, from the year 2002, used very simple equations to estimate piezoelectric power, which are now known to be incorrect. The authors need to re-establish the analytical equations. Authors response: The provided equation focuses on mechanical stress and its conversion into electrical power in piezoelectric transformer. It describes the output power in terms of the mechanical properties of the piezoelectric material. However, the following equation P = (g33*F*h)2/R; relates to the electrical characteristics of the piezoelectric material and focuses on the power generated in terms of electrical voltage and load resistance in piezoelectric generator. Both are correct. We kindly request that the respected reviewer clearly differentiate between the operation of the piezoelectric device as a generator and as a transformer. If the respected reviewer is still requesting the reestablishment of the equation, before we proceed with the revision, we kindly ask that the he/she clearly specify the equation they are referring to, along with the relevant reference.

Reviewer 3 Report

Comments and Suggestions for Authors

This a paper about  radio frequency (RF) energy harvesting. Good introduction, adequate scope of literature review & citation. All other sections, like current approach, experimental setup, measurement and analysis, benchmarking with SoTA are satisfactory. Comparison with other reported in the literature data is provided. Conclusions are correct. Figures and in generally paper is prepared in reasonable form. No negative comments.

Author Response

Response to the Comments and Suggestions from Reviewer 3:

This a paper about radio frequency (RF) energy harvesting. Good introduction, adequate scope of literature review & citation. All other sections, like current approach, experimental setup, measurement and analysis, benchmarking with SoTA are satisfactory. Comparison with other reported in the literature data is provided. Conclusions are correct. Figures and in generally paper is prepared in reasonable form. No negative comments. Authors response: Thank you very much for your thorough and positive review of our paper. We truly appreciate your valuable time and feedback.

Round 2

Reviewer 1 Report

Comments and Suggestions for Authors

Authors have well addressed my previous comments.

Author Response

Thank you very much for accepting our manuscript